# Highly Aggressive Osteosarcoma of the Ethmoids and Maxillary Sinus-A Case of Successful Surgery and Proton Beam Radiotherapy in a 65-Year-Old Man

**DOI:** 10.3390/medicina58091141

**Published:** 2022-08-23

**Authors:** Jaromír Astl, Tomas Belsan, Ludmila Michnova, Jiří Kubeš, Tomas Filipovsky, Jiri Blecha, Richard Holy

**Affiliations:** 1Department of Otorhinolaryngology and Maxillofacial Surgery, Military University Hospital, 16902 Prague, Czech Republic; 2Third Faculty of Medicine, Charles University, 10000 Prague, Czech Republic; 3Department of Otolaryngology, Institute of Postgradual Medical Education, 10005 Prague, Czech Republic; 4Department of Radiology, Military University Hospital, 16902 Prague, Czech Republic; 5Department of Pathology, Military University Hospital, 16902 Prague, Czech Republic; 6Proton Therapy Center Czech, 18000 Prague, Czech Republic; 7Department of Oncology, Second Faculty of Medicine, Charles University, University Hospital in Motol, 15000 Prague, Czech Republic

**Keywords:** ethmoid bone, maxillary bone, oncology, osteosarcoma, prognosis, protocol, survival data, treatment

## Abstract

Sarcomas in the head and neck area are rare diseases with an incidence of under 1% of all head and neck malignant tumours. Osteosarcomas or osteogenic sarcomas consist of neoplastic cells that produce osteoid bone or immature bone. Sarcomas develop more in the mandible than the maxilla. The exact diagnosis of different types of sarcomas is based on the immunohistochemical investigation. These rare tumours are of mesenchymal origin; osteosarcomas and chondrosarcomas are the most common types—Ewing’s sarcomas. The use of proton beam radiotherapy in the treatment of osteosarcoma of the maxilla is rarely reported in the literature. We present a case of successfully treated (surgery and proton beam radiotherapy) poorly differentiated highly aggressive osteosarcoma in the ethmoids and maxillary sinus and morbidity after the treatment. We were presented with a case of a 65-year-old man with anaesthesia and palsy of the right face. The stomatology department performed the extraction of a tooth. One month later, the wound was still open. The histology showed an osteogenic sarcoma in the area of the wound. The oncologist and maxillofacial surgeons in a catchment hospital recommended a nonsurgical approach. Hence, we performed a radical maxillectomy and ethmoidectomy, after which we continued with proton bean radiotherapy. The patient is now five years after therapy without signs of sarcoma; however, he has blindness in the right eye and reduced vision in the left eye, as well as gliosis of the brain, vertigo, tinnitus, trismus, and ancylostomiases. Head and neck osteosarcomas treatment is considered a complex multidisciplinary task. It is currently argued that there is no clear therapeutic protocol for successful treatment. Innovations in treatment modalities such as proton beam radiotherapy appear to have potential, although their effects on long-term morbidity and survival outcomes are still undetermined. We present a rare case report of an osteosarcoma of the maxilla involving an innovative, successful treatment procedure combining surgical excision followed by proton beam radiotherapy. This treatment approach may enable maximum tumour control. This protocol has not been adequately described in the world literature for this diagnosis.

## 1. Introduction

Osteosarcoma (also called osteogenic sarcoma) is a malignant tumour composed of osteoid bone or immature bone cells. Osteosarcomas account for less than 7% of all bone sarcomas. In the head and neck area we can find fibrosarcomas, chondrosarcomas, and osteosarcomas as frequent sarcomas, and more rarely leiomyosarcomas, liposarcomas, and rhabdomyosarcomas. The exact diagnosis of different types of sarcomas is based on the histology and immunohistochemical investigation of the expression of different proteins. These rare tumours are of mesenchymal origin; osteosarcomas and chondrosarcomas are the most common types—Ewing’s sarcoma. Compared to the clinical presentation of osteosarcomas of the long bones, osteosarcomas of the jaw occcur in an older patient population (30–50 years). The mandible is more often affected than the maxilla [1].

The Enneking classification with UICC is used for osteosarcomas (7th ed., 2011) [2]. The recommendation for large bone resection or radical bone resection depends on the stadium, low- or high-risk sarcoma type, metastases, and TNM status [3].

The treatment of the sarcoma is based on radical surgery, irradiation of the tumour origin and/or lymph nodes, and chemotherapy if needed. The treatment of sarcomas is difficult, as there are only moderate data about the survival of the patients and disease-free interval. The determination of treatment depends on the exact diagnosis, large-cell typing of the sarcomas, and the low incidence of these patients in the population [4]. The survival data of the sarcoma patients are different and depend on the type of sarcoma.

The use of proton beam radiotherapy in the treatment of osteosarcoma of the maxilla is rarely reported in the literature.

The moderate quality of life with the radical therapy of sarcomas is described in literature, too. The treatment of sarcomas is difficult in clinical practice because of its low frequency in the population. Many papers refer to low numbers of treated patients and/ or described case reports only [5]. Early diagnosis and treatment are essential to improve long-term prognosis [6]. 

We present a case of successfully treated (surgery and proton beam radiotherapy) poorly differentiated highly aggressive osteosarcoma in the ethmoids and maxillary sinus with survival data, disease-free interval, local recurrence, metastases, and morbidity of one patient. This patient has survived over 5 years—overall survival interval (OS). This case report describes the morbidity and complications of the surgery and proton beam therapy. The patient signed an informed consent and gave his consent for publication.

## 2. Case Report (Observation)

A 65-year-old man (ethnic group—Western Slavic origin) was referred to the Otorhinolaryngology outpatient clinic of the Department of Otorhinolaryngology and Maxillofacial Surgery in the Military University Hospital Prague, 3rd Faculty of Medicine of the Charles University Prague with 3 months of difficulty swallowing and pain in the right face that was treated by extraction of tooth 18 (only). He did not smoke and consumed a small amount of alcohol—200 mL wine each second day.

A tumour mass was found in the gingival arch where tooth 18 was extracted. The CT scan showed an enlargement of the tumour mass spread into the maxillary bone (see Figure 1 and Figure 2.)

The maxillary sinus was completely filled by the tumour mass, which spread into the floor of the orbit and lateral nasal wall. A sample was taken from the tumour mass by the stomatology outpatient department. A histopathological diagnosis of an undifferentiated sarcoma was made. 

Radical surgery was the next step in the treatment protocol. We performed a total maxillectomy with partial orbital resection, resection of the zygomatic complex, ethmoidectomy, and complete resection of medial nasal wall with conchae on the right side. The extension of the surgical radicality was focused on the radical resection of the whole tumour tissue with an R0 margin. The wound was closed by a face flap. No surgery was performed on the neck lymph nodes, because there were no suspicious lymph nodes detected by imagining methods. 

The histological analysis revealed a sarcoma with detection of a fascicular or storiform structure of cells. The cells were irregular and hyperchromic with multiple and/or polymorphic nuclei.

There were also large atypic cells associated with high mitotic activity and atypia of mitotic activity (25/10HUF). These cells are highlighted by black arrows in Figure 3.

The neoplastic cells did not express desmin, MSA, or CD34; the S-100 protein was positive in arterials including CD31. The immunobiology for MDM2 and CDK4 were positive. This sign was typical for liposarcomas and osteosarcomas. Hence, the diagnosis of a poorly differentiated highly aggressive osteosarcoma was made (see Figure 4).

One month later, the patient had a recurrence of the lesion in the ethmoid region, with a size of 4 × 5 × 6 mm. After a multidisciplinary discussion made by an oncologist, otorinolaryngologist and radiation oncologist, the resection of this tumour was performed by an endoscopic approach, and external proton beam radiotherapy was used post-surgically.

Proton Beam Radiotherapy

Eight weeks after the oncological surgery, proton beam radiotherapy was performed in the Proton Therapy Centre Czech Republic in Prague. 

Full doses of proton beam therapy were applied to the region of the tumour, with a focus on the maxilla and ethmoidal part on the right side. The right eyeball was spared during the therapy. A skin defect in the face flap developed during the therapy, which caused prolonged healing for the following seven weeks. 

The target volume was the paranasal sinuses tumour right side + rim. The dose applied (CGE) was 70 Gy, with 35 fractions, and the dose per fraction (CGE) was 2 Gy.

The radiation technique was proton therapy using the pencil beam scanning technique (See Figure 5).

In terms of the organs at risk-dose: glandula parotis right side D mean 4.89 Gy, cochlea right side D mean 1.4 Gy, nervus opticus right side D max 48.4 Gy, nervus opticus left side D max 41.7 Gy, chiasma D max 47.4 Gy, retina right side D max 47.8 Gy, retina left side D max 30.6 Gy, brainstem D max 4.1 Gy, spinal cord D max 0.3 Gy, right eye 48.6 Gy, and left eye 31.6 Gy.

Treatment was included for advisory dermatitis grade 2, stomatitis grade 2, soor, and conjunctivitis grade 1.

We observed the patient for the next five years. Two years after the start of the therapy, the right eye was blind. 

Five years later, the patient presented with a decline in the function of his left eye. The patient showed structural changes in the brain on the left side-the MRI showed large gliotic parts in the frontal and parietal part of the brain (see Figure 6 and Figure 7).

Ancylostomiases and trismus started in the first year following the proton therapy. The patient could open his mouth 6–7 mm only, which influenced his nutritional status. The patient ingested minced food and beverages. However, his weight remained stable for five years.

Postoperatively, the patient was referred from the otorhinolaryngologic surgeon to the oncologist. Regular MRI follow-up examinations were performed every 6 months. A CT scan of the lungs was performed once a year. A PET-CT was indicated in the year 2021 (4 years after proton beam radiotherapy). None of the images revealed any evidence of locoregional cancer recurrence or metastatic involvement. 

No local recurrence was observed (OL 0); the overall survival time has been 6 years as of the date of publication (OS 6 years); the disease free interval has been 5 years and 6 months, and no metastases were observed (MT 0).

## 3. Discussion

Many case reports of osteosarcomas of the jaws are described, including recommendations for diagnostic and therapeutic management. The main reason for this report was the successful treatment of osteosarcoma with surgery followed by proton beam radiotherapy. The use of proton beam radiotherapy in the treatment of osteosarcomas of the maxilla is not adequately described in the literature.

Osteosarcomas in the head and neck region are rare, difficult to diagnose, and may present with varying signs and symptoms [6]. Osteosarcomas are difficult to diagnose because of false biopsy results which range from 17% to 25% [3,4]. The essential histological characteristic of osteosarcomas is the existence within a sarcomatoid stroma of skeletal tissue arranged in an anarchic manner and synthesized by atypical osteoblasts, having malignant characteristics. Well-differentiated osteosarcomas present a challenge for histological analysis, because they are often confused with benign fibro-osseous lesions such as fibrous dysplasia or ossifying fibroma [7,8].

Maxillary osteosarcomas usually develop in people between 20 and 40 years of age. In our case report, the patient was a 65-year-old man. In contrast, the case report of French authors described a case of osteosarcoma of the maxilla of a 17-year-old patient [1,9].

Due to the anatomical location of maxillary osteosarcomas, dentists are the first to assess the lesion in 45% of cases. Unfortunately, tooth extraction occurs in two-thirds of cases [5]. The most common complaint is painless swelling, which can sometimes be associated with dysesthesia and/or limited mouth opening [1]. These lesions in the maxilla may mimic a periapical lesion of endodontic (dental) origin, and the differential diagnosis is mandatory [6].

CT and/or magnetic resonance imaging are recommended for maxillary tumours. These imaging methods are necessary to assess the exact description of the spread of the tumour into the maxillary bone, ethmoids, and infratemporal fossa. These facts are important for planning the resection of a tumour as well as for reconstruction. Diagnostic ultrasound can be used for the lymph nodes on the neck, which can be associated with the tumour only. The CT and MRI criteria typical for osteosarcoma were summarized by Luo et al. [10].

Chen et al. described a prognostic factor for osteosarcoma. They discussed that histological grade and negative margins were significant factors affecting the survival of patients [11]. For sarcomas, statistically important factors for survival of the patients have been identified: the local recurrence (LR), overall survival (OS), disease-free survival (DFS), and metastasis (MT).

Many authors recommend a diagnostic set including:Clinical investigation of all parts of the head and neck (endoscopy to provide evidence of tumour).A tissue sample can be taken, if it is a safe procedure.CT, MRI scan (size, spread, and metastases of tumour mass), if necessary, PET scan, and USG.Histopathology: complete investigation and extensive immunohistochemistry tests are necessary.

The treatment of osteosarcomas combines two modalities: surgery and beam therapy [10,12]. 

Laskar et al. evaluated the efficacy and toxicity of dose-escalated IGI modulated radiation therapy in osteosarcoma. The five-year local control improved the survival in 66% [1,13].

A database collected 821 patients with osteosarcoma (HNOS) treated by neoadjuvant chemotherapy and radical surgery and adjuvant chemotherapy. The authors described no benefit in perioperative chemotherapy or radiation therapy [14].

However, Brady et al. found a higher 5-year survival rate in paediatric patients with surgery treatment compared to radiotherapy alone [15].

Kalvarezos and Sinha [16] provided an overview of the treatment of sarcoma, osteosarcomas included. Surgery is the main modality in the treatment protocol in low-grade tumours. For high-grade tumours, neoadjuvant chemotherapy followed by surgery and radiotherapy was recommended. As the bone sarcomas can be radioresistant, the role of proton beam therapy was discussed.

Sufficient data for proton beam radiotherapy currently do not exist. That is why we added our experience with the treatment of osteosarcoma in our patient. These data can improve the knowledge of proton beam therapy’s effect, namely good survival data and a moderate quality of life. In the literature, the use of proton beam therapy for maxillary sarcomas has been described in only one case report- Ewing sarcoma family tumours-from Japan, where a 4-year-old patient was treated with chemotherapy followed by partial maxillectomy. After surgery, proton beam radiotherapy was performed. Partial resection combined with proton beam therapy may enable maximal tumour control and minimal functional and cosmetic side effects [17].

Netherlands authors mentioned that they did not use chemotherapy for their patient.

The reason for that was the moderate data on its effect on osteosarcomas [18]. The study showed that (neo-) adjuvant therapy was associated with a smaller risk of local recurrence in patients under 75 years old. Still, renal and hepatic function should be assessed prior the usage of chemotherapy [18].

Treatment:Surgery: surgical therapy must be radical; the surgical margins are very important for grading of sarcomas in general.Radiotherapy: beam therapy (photon, proton, or isotope) depends on staging, including grading, histological type of sarcoma, and the radicality of surgery when performed.Chemotherapy: for aggressive histological subtypes of osteosarcoma, chemotherapy is necessary in many cases, a combination of vincristine, doxorubicin, cyclophosphamide, and etoposide have been used.

Follow-up for sarcoma:A clinical report including endoscopic methods, Narrow band imaging endoscopy (NBI) and Ultrasonography (USG) every 4 months;MRI, once yearly (PET scan if necessary).

Some authors have stated that there is no standardized staging system for soft-tissue sarcomas of the head and neck area [19,20]. So far, the UICC classification [19] classifies soft-tissue sarcomas (different to rhabdomyosarcoma) in two levels. Osteosarcomas are level I in this classification. To this day, we do not have data for personal tailoring of the therapy of osteosarcomas. For the head and neck region, the soft-tissue sarcoma classification is used. [19,20] The sarcoma of the head and neck region do not have any division into the stadium of disease. Radical therapy is necessary in the treatment of osteosarcomas in accordance with individual quality of life results [21,22]. 

Current knowledge of osteosarcoma of maxilla:

Osteosarcoma of the maxilla is a rare disease, that is mainly reported in the literature in the form of single case reports. 

Diagnosing osteosarcoma is very challenging and osteosarcoma is often misdiagnosed as osteomyelitis due to the nonspecificity of its symptoms upon initial presentation [6,23,24].

The vast majority of authors emphasise the importance of surgical resection of the lesion with clean margins.

The management of head and neck osteosarcomas comprises a multi-disciplinary approach [24,25].

Portuguese authors have reported that surgical resection of a lesion with clear margins followed by chemotherapy is associated with the best prognosis for osteosarcoma of the jaw [6].

Indian authors have discussed that osteosarcomas of maxillofacial region pose difficulties in obtaining tumour-free margins due to their complex anatomy and close proximity to the cranium. Surgery may be complemented by radiotherapy with or without chemotherapy [24]. 

An American retrospective review of the literature on sinonasal sarcomas (over the last 30 years) reported a significant association between overall stage and overall survival on univariate and multivariate analysis, and overall stage appeared to be the most significant contributor to survival. Surgical excision with negative margins, when possible, combined with adjuvant radiation and/or chemotherapy, appears to offer the best survival outcomes [25].

Japanese authors have reported the treatment protocol in a case report (Ewing sarcoma of maxillary sinus, 4-year-old girl): chemotherapy followed by partial maxillectomy and after surgery, proton beam radiotherapy was performed [17]. 

Authors from the United Kingdom have reported that heavy-particle therapy (proton beam) in combination with surgery is increasingly being used to treat otherwise unresectable diseases, particularly in children [16].

The role of proton beam radiotherapy is promising, but to our knowledge no long-term data currently exist [16].

Increasing studies have proved that the imbalance of the immune system and tumour immunity is a vital factor leading to the pathogenesis and deterioration of osteosarcoma [26,27].

It is currently argued that there is no clear therapeutic protocol for successful treatment, and that research into the pathogenesis and underlying molecular mechanisms of osteosarcoma is necessary to achieve new breakthroughs in osteosarcoma therapy [26,27].

A new study from China shows that CBR3-AS1 plays an oncogenic role in osteosarcoma through the modulation of the miR-140-5p/DDX54-NUCKS1-mTOR signalling pathway network [26].

## 4. Conclusions

Head and neck osteosarcomas treatment is known as a complex multidisciplinary task. 

It is currently argued that there is no clear therapeutic protocol for successful treatment [26,27].

Surgical excision with negative margins when possible combined with adjuvant radiation and/or chemotherapy appears to offer the best survival outcomes [6,17,24,25].

Innovations in treatment modalities such as proton beam radiotherapy appear to have potential, although their effects on long-term morbidity and survival outcomes are still undetermined.

We presented a rare case report of an osteosarcoma of the maxilla involving an innovative successful treatment procedure combining surgical excision followed by proton beam radiotherapy. This treatment approach may enable maximum tumour control. This protocol has not been adequately described in the literature for this diagnosis. 

Still, the radical treatment protocol is associated with comorbidities and decreasing quality of life. 

## Figures and Tables

**Figure 1 medicina-58-01141-f001:**
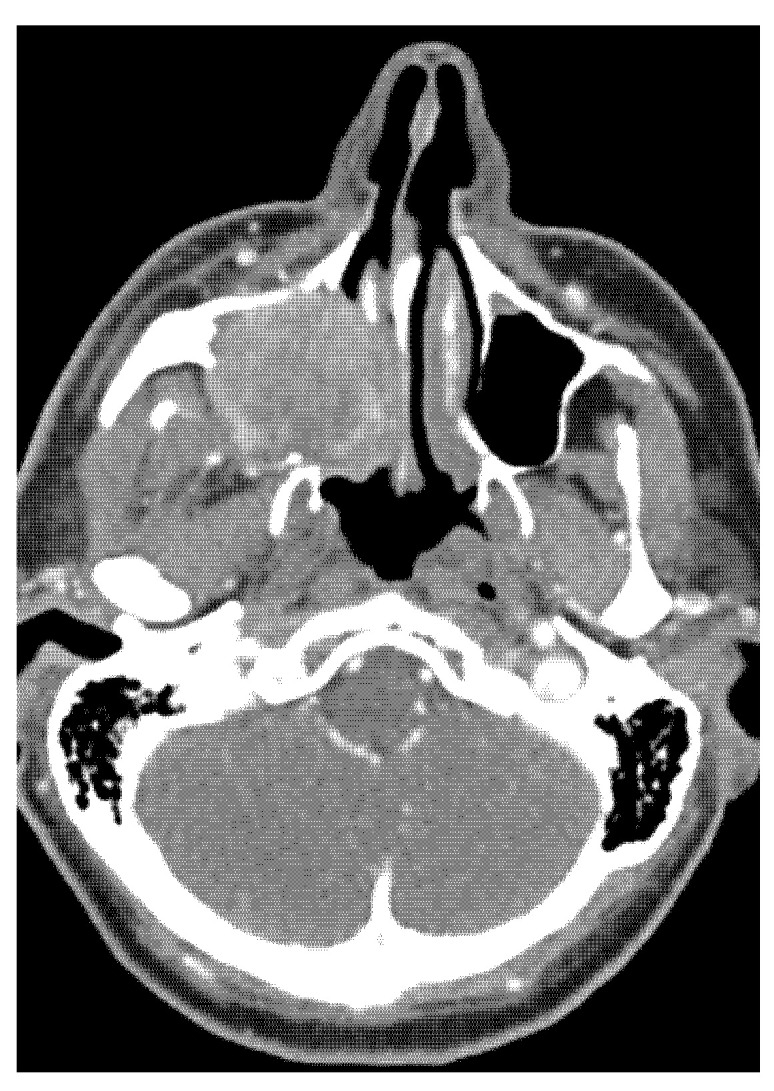
Axial CT imaging demonstrates a solid nonhomogeneous tumour that completely fills the right maxillary sinus, destroying the medial and dorsolateral wall of the sinus and the base of the right orbit. The CT images are published with the permission of the Radiology Department, The Hospital of České Budějovice, Czech Republic.

**Figure 2 medicina-58-01141-f002:**
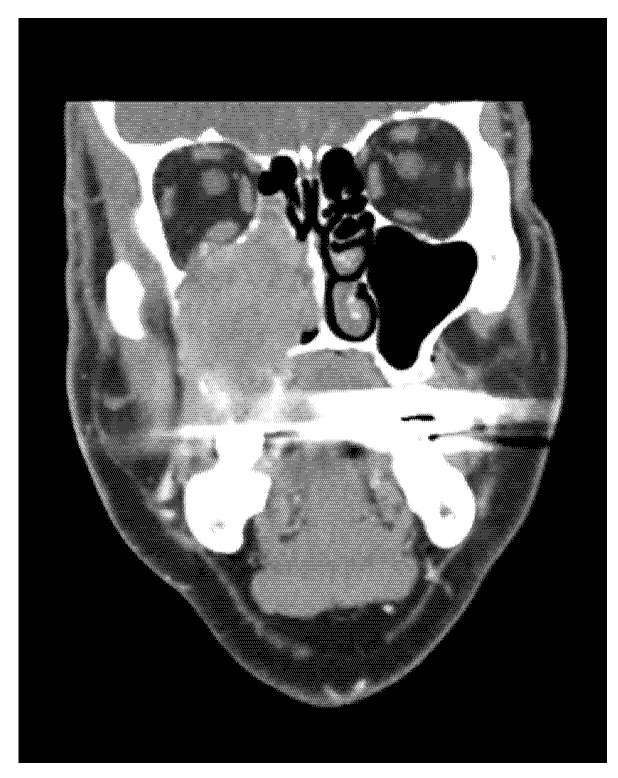
Coronal CT imaging demonstrates a solid nonhomogeneous tumour that completely fills the right maxillary sinus, destroying the medial and dorsolateral wall of the sinus and the base of the right orbit. The CT images are published with the permission of the Radiology Department, The Hospital of České Budějovice, Czech Republic.

**Figure 3 medicina-58-01141-f003:**
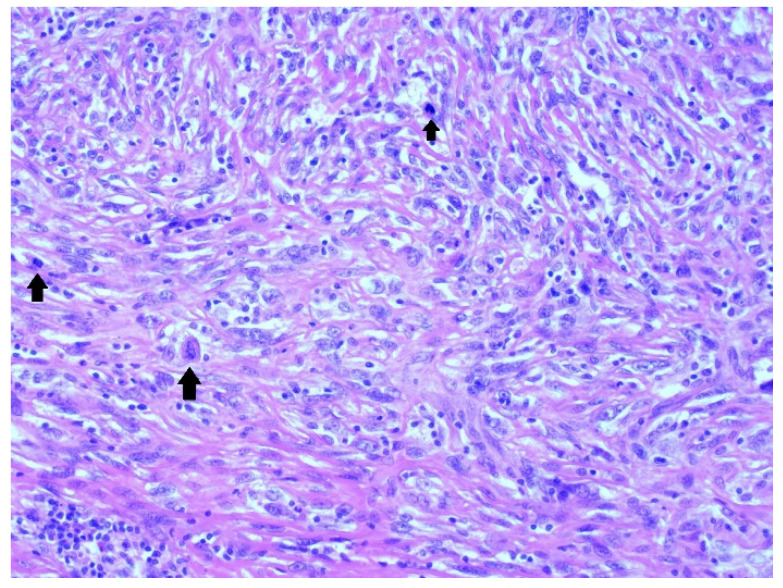
Highly cellular undifferentiated sarcoma in a fascicular pattern with increased mitotic activity—up arrows (H&E staining, 200×).

**Figure 4 medicina-58-01141-f004:**
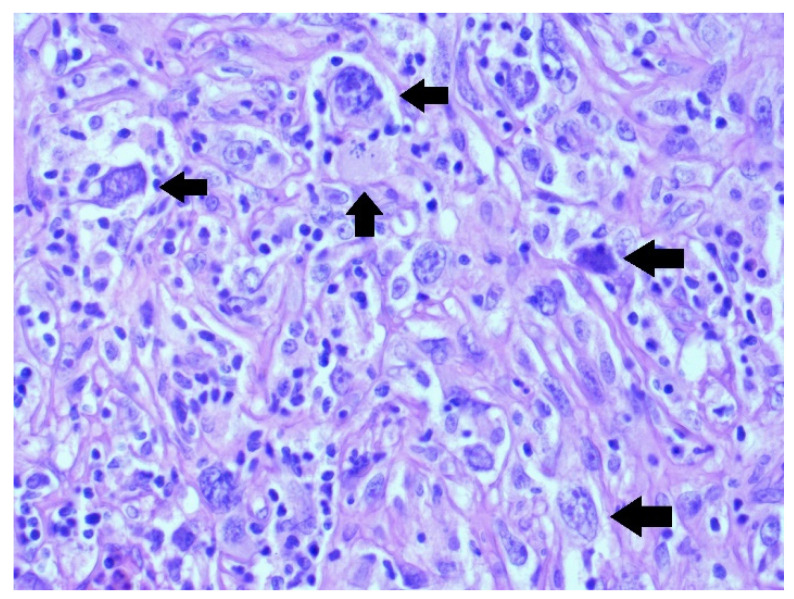
Pleomorphic tumour cells with enlarged round to bizarre nuclei—side arrow; atypical mitotic figure—up arrow (H&E staining, 400×).

**Figure 5 medicina-58-01141-f005:**
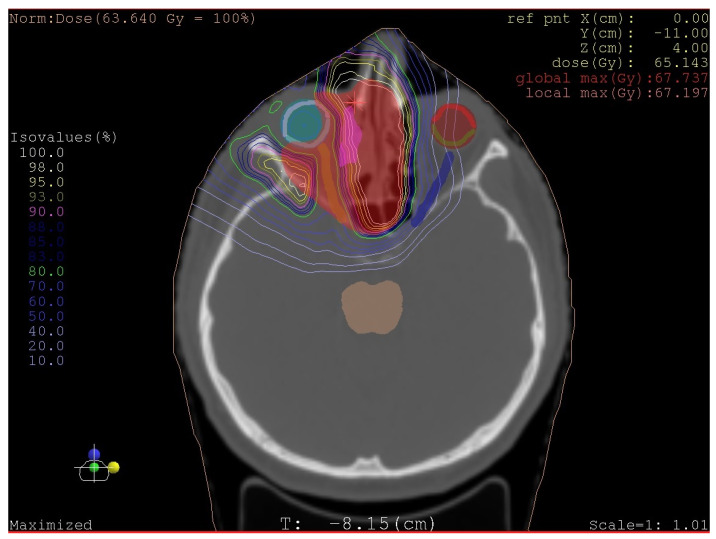
Plan Proton beam radiotherapy, Isodose Slice View.

**Figure 6 medicina-58-01141-f006:**
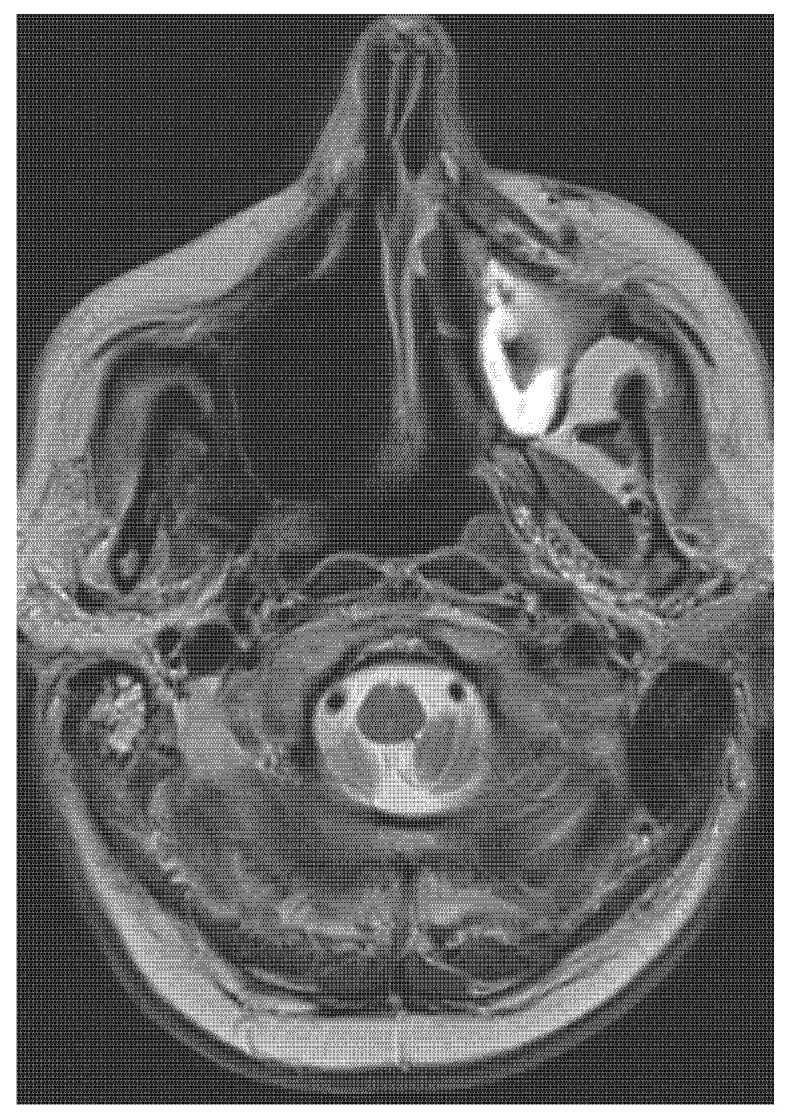
Follow-up MR imaging demonstrates a postresection cavity without tumour residue or recurrence in the right maxilla (5 years after surgical treatment). Axial imaging in T2 weighting.

**Figure 7 medicina-58-01141-f007:**
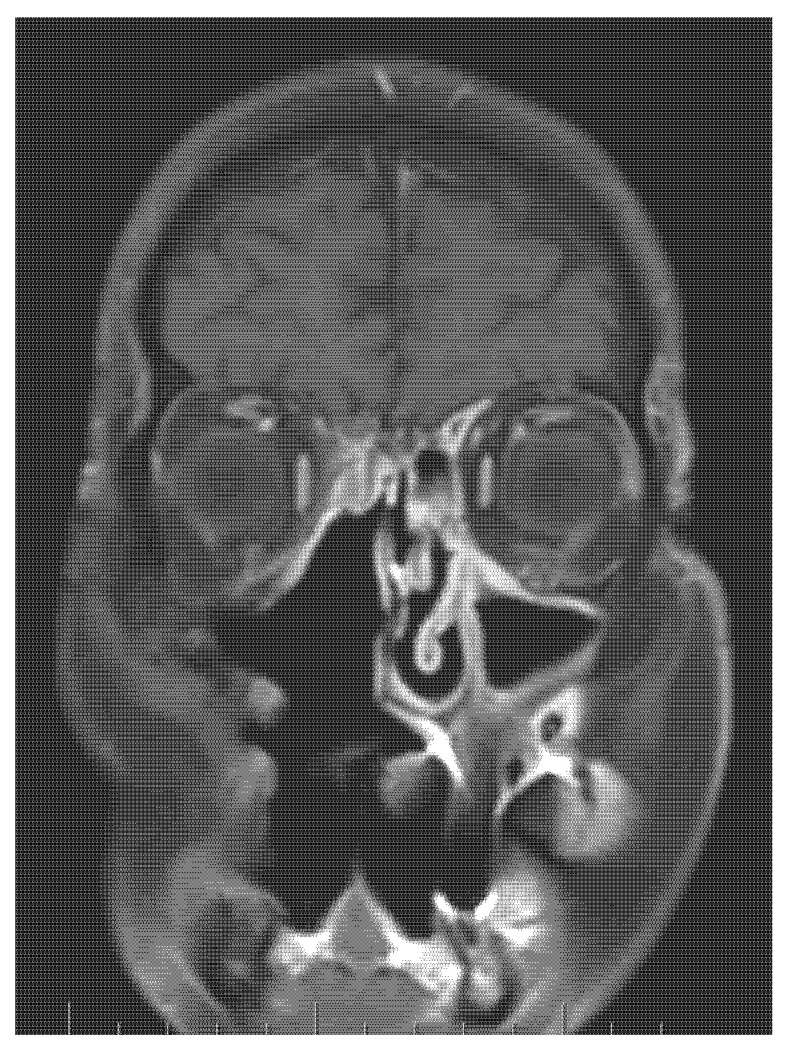
Follow-up MR imaging demonstrates a postresection cavity without tumour residue or recurrence in the right maxilla (5 years after surgical treatment). Coronal imaging in T1 weighting after contrast agent administration.

## Data Availability

Not applicable.

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
