# Peer review of "Highly Aggressive Osteosarcoma of the Ethmoids and Maxillary Sinus-A Case of Successful Surgery and Proton Beam Radiotherapy in a 65-Year-Old Man"

_medicina, 2022, doi:10.3390/medicina58091141_

Round 1
Reviewer 1 Report
The authors have preswented a paper about "High aggressive osteosarcoma of ethmoid and maxillary sinuses a case of successful surgery and proton beam treated patient".
The topic is not new but still interesting.
I have some major concerns about the paper:
1) Why did not the authors use also PET-CT? it could provide more details in such an advanced case also for sutemic staging (M)
2) In figure 2 there is clearly an artifact but no mention is made by the authors: how could they prepare an accurate presurgical plan with such low quality imaging?
3) There are some typos all around the mansucript which should be edited
4) No mention is provided with regard to the consent by patient to publish this material: do the authors have a written informed consent? can they provide it?
5) The discussion in poor and it should be much improved
6) was this case discussued in a multidisciplianry setting? it is not clearly stated
7) it would be an affition to inclued a review of literature to the case report
Author Response
Attached is a reply for the reviewer.

Reviewer 2 Report
Dear authors, I am sending you a few concerns.
The Title and Abstract look acceptable.
I recommend the keywords to be placed by alphabetic order, and I suggest the term “oncology” in order to increase the paper reach.
The manuscript visual format is not according to the journal template.
In the Introduction, when the authors mention the low incidence rate I believe it would be welcome com give a few more data and numbers. What are the incidence rate of osteosarcoma in the jaws after all. Additionally, I believe it would be very welcome to mention/debate that these lesions in the maxilla may mimic very closely a periapical lesion of endodontic (dental) origin and the differential diagnosis is mandatory. In this aspect I suggest the manuscript DOI: 10.1111/aej.12491 to be mentioned and support that mention/debate.
May the authors provide the patient ethnic group?
Is it possible to have the medical/dental frame that lead the tooth 18 to be extracted? The tooth was only extracted or any enucleation was attempted? Is it possible to have a better frame from those previous events?
Please change the “pbserved” for “observed”. Please check any possible grammar or syntax mistakes.
Is it possible to place Figures 1 and 2 side by side instead top-bottom?
In the Discussion it would be welcome the major points that may support the choice of one treatment over the others. I would it is case-dependent, but I say in major global terms.
Please notice the that reference list is not in accordance to the journal guidelines.
Author Response
Attached is a reply for the reviewer.

Reviewer 3 Report
This case report is aimed to present the successful management of osteosarcoma using a combination of therapies.
Despite some novelty in the use of proton beam therapy for maxillary osteosarcoma, the manuscript is hard to read. It is strongly advised to re-write the entire manuscript using the CARE statement and a language editing service.
The introduction fails to support the relevance of this report, particularly the use of proton beam therapy, which seems to has a central role in the management of this patient.
The case description needs to be re-arranged and includes clinical, biological, or other relevant data that could explain the success in this case.
Discussion and conclusion must focus on the case, and what the case report adds to the existing literature on this topic.
Author Response
Attached is a reply for the reviewer.

Round 2
Reviewer 1 Report
Please correct line 104 the word "radiologist" with "radiation oncologist"
Author Response
Dear Reviewer,
the line 104 was corrected : to the word "radiologist" with "radiation oncologist"
Reviewer 2 Report
Dear author, I have no more concerns.
Author Response
Dear Reviewer ,
Thank You.
Reviewer 3 Report
This new version of the manuscript has noticeable improvements in grammar and readability. However, not much improvement is seen in the presentation of data and facts along the text.
- The introduction now included some additional data, but it is still failing to provide a rationale for this report.
- The case description also added some minor new data, which appears useful. But the order of presentation must improve.
- Discussion, as expected, included some new papers on the matter. But most of this section is focused on describing the existing literature as a bullet list and repeating some info provided in the Introduction. It fails to support the relevance of this paper.
- The conclusion must be rewritten and focus on a clear statement of what this paper adds to existing literature.
Author Response
Dear Reviewer,
the answer is attached.
